# The impact of anthropometric characteristics on isometric trunk muscle endurance tests: A reliability and performance analysis

Casto Juan-Recio[1], Francisco J. Vera-Garcia[1,2]\*, Alejandro Lopez-Valenciano[3], David Barbado[1,2]

1 Sports Research Centre, Department of Sport Sciences, Miguel Hernández University of Elche, Avda. de la Universidad s/n, Alicante, Spain, 2 Institute for Health and Biomedical Research (ISABIAL Foundation), Miguel Hernández University of Elche, Avda Pintor Baeza, 12—Planta 5a Centro de Diagnóstico, Alicante, Spain, 3 Department of Education Science, School of Humanities and Communication Sciences, Universidad Cardenal Herrera-CEU, CEU Universities, Calle Grecia 31, Castellon de la Plana, Spain

\* fvera@umh.es

## Abstract

The Biering-Sorensen test (BST), the Side Bridge test (SBT) and the Ito test (IT) are three of the most used field-based tests to assess isometric trunk muscle endurance. The objectives were to analyze the relationship between the participants' test performance and several anthropometry parameters, the data consistency and the sex effect on test scores. Forty-five recreational athletes (27 males and 18 females) performed the three isometric trunk holding tests twice in two testing sessions to perform the reliability analysis and later, the three tests were performed once more, but in different sessions (one for each test) to maximize test performance and reduce the bias of muscle fatigue. Data (i.e., test scores and anthropometric variables) were logarithmic transformed to ensure the normality and homoscedasticity assumption. Relative reliability was very good, with ICCs > 0.70 in all tests, while absolute reliability showed high values of typical error (12.1–24.1%). ANOVA showed significant differences between sessions for the BST and the SBT scores and between sexes in the BST (females: $193.7 \pm 53.2$ s; males: $161.9 \pm 52.2$ s). IT scores showed a negative correlation with mass in both sexes (males: $r = -.436$; $p = .026$; females: $r = -.562$; $p = .019$) and with biileocrestal breadth ($r = -.735$; $p = .001$) and biacromial breadth ($r = -.745$; $p = .001$) in females. BST scores correlated significantly with biacromial breadth ($r = -.379$; $p = .050$) in males. SBT scores were negatively correlated to mass ($r = -.703$; $p < .001$), biileocrestal breadth ($r = -.672$; $p < .001$), biacromial breadth ($r = -.601$; $p = .001$) and acromion-iliac index ($r = -.493$; $p = .010$) in males and to relative lower extremity length ($r = -.493$; $p = .038$) in females. In conclusion, trainers and clinicians should consider individual anthropometric and sex differences when interpreting test results, as larger body mass and upper body breadth may artificially lower endurance scores. Adjustments to normative values may be required in applied

**Data availability statement:** All relevant data are within the manuscript and its Supporting Information files.

**Funding:** This research has been funded by the R+D+i for emerging research groups 2023 project of the Consellería de Educación, Universidades, y Empleo: CIGE/2022/ 22

**Competing interests:** NO authors have competing interests

settings. Moreover, based on the reliability analysis, these tests could be used to classify participants consistently, but the BST and the SBT require an extensive familiarization period and they don't seem to be useful to detect small changes in participants' performance over time.

## Introduction

Considering the fact that trunk muscle endurance deficits and imbalances have been related with low back pain [1–3] and that trunk muscle fatigue has a harmful effect on muscular coordination, postural control and spine stability [4,5], the development of trunk muscle endurance is a common training program goal for injury treatment and prevention, sport performance and functional capacity improvement in daily tasks [6–8].

Isometric trunk muscle endurance field-based tests, as the Biering-Sorensen test, the Plank test, the Side Bridge test, etc., are some of the most widely used tests to assess participants´ trunk muscle endurance and progress in sport, education, research and clinical settings [6,9–11]. These tests basically consist of holding different body parts (especially the trunk) suspended against gravity for as long as possible, and therefore they are easy to use and do not need expensive materials nor a complex data analysis [10]. However, bearing in mind that the participants' anthropometric characteristics (e.g., mass, width, and length of the body segments held against gravity) have a great influence on the gravitational torque that the participants must counteract to maintain the position, these may have a significant impact on the test results. In this sense, some studies have found significant negative correlations between participants´ mass [12–17] or body mass index [18,19] and different isometric trunk holding tests (i.e., Biering-Sorensen test, Plank test, Side Bridge test, etc.) in different populations (i.e., baseball players, firefighters, healthy people, college students, etc). Moreover, Dejanovic et al. [20] also found slight significant correlations between different anthropometric measures (i.e., biacromial breadth, sitting height, etc.) and the Biering-Sorensen test, the 60º Flexion test, and the Right and Left Side Bridge test in children between 7 and 14 years of age. Despite these correlations, which suggest that an individual's body mass and its distribution may have an impact on the isometric trunk muscle endurance test performance [9,16], the participants' anthropometric characteristics are not usually taken into consideration when using these protocols (e.g., to obtain trunk muscle endurance normative data) [9,11,15,21–23]. Further research is needed to better understand the effect of anthropometric characteristics on these test scores, including which anthropometric variables have a greater impact on test performance and the potential impact of sex-based differences on test scores.

In order to answer these questions and facilitate the use and interpretation of some of the most representative isometric trunk muscle endurance field tests [i.e., Biering-Sorensen test (BST), Abdominal Ito test (IT), Side Bridge test (SBT)], the main objective of this study was to analyse the relationship between these test

scores and several anthropometric measurements in young physically active males and females. The sex effect was analysed to facilitate the test score interpretation and application in different populations. In addition, absolute and relative reliability was also calculated to provide valuable information about the characteristics of the referred tests and to avoid the potential bias produced by inconsistent variables on correlations analyses [24]. Based on previous studies [12–20], we hypothesize that individuals with greater body mass and broader anthropometric dimensions will exhibit lower endurance test performance due to increased gravitational torque. In addition, sex-based differences will influence test performance, with females potentially showing higher endurance in certain tests due to differences in muscle fiber composition and fat distribution. These findings could improve the use and interpretation of these trunk muscle endurance field tests both in clinical and in sport settings.

## Methods

### Participants

A total of 45 young healthy volunteers (27 males: 23.5±4.0 years, 75.7±10.3 kg, 177.4±7.2 cm; 18 females: 25.0±6.7 years, 62.7±6.4 kg, 166.1±3.1 cm) were recruited from a university student population. They were recreational athletes (i.e., practicing soccer, gymnastics, basketball, running, etc.), with a 60–120 min per day moderate to intense physical activity regular practice 2–5 times a week (for a total of 120–300 min per week, approximately). They completed a questionnaire about their medical and athletic history to evaluate their health status and regular physical activity. The inclusion criteria for this study were: not having any known medical problem or episode of low back pain in the six months prior to the study and not to be participating in any structured trunk exercise program at the time of the research. The participants were instructed not to modify their level of physical activity during the time of the study (especially in relation to the trunk muscle conditioning) nor to perform vigorous physical activity during the 24 hours before the testing sessions. Before starting the data collection, participants were informed of the study risks and aims. Participants filled out and signed a written informed consent in accordance with the 2013 Declaration of Helsinki. All procedures were approved by the University Research Ethics Committee (DCD.CJR.230630).

### Procedures

Following a testing schedule, each of the participants performed five testing sessions separated by 1-week periods. In the first and second testing session, the three isometric trunk muscle endurance tests were performed in a counterbalanced way with an 8-min recovery period between each [9,11,25] to perform a test-retest reliability analysis. Then, in the third, fourth and fifth testing session, the tests were performed separately (i.e., a test in each session, to avoid the fatigue effect caused by one test on another) to analyze the relationship between test scores and anthropometric variables. The participants did not perform any familiarization session before the testing sessions which made the evaluation of the learning effect of the three trunk tests possible.

**Isometric trunk muscle endurance tests.**  The IT [26], the BST [27], and the right SBT [11] were performed to asses flexor, extensor, and lateral flexor trunk endurance, respectively as described by Juan-Recio et al., [10]. These trunk endurance tests basically consist of holding the trunk against gravity for as long as possible in a flexed, prone horizontal, and lateral position, respectively (Fig 1).

**Anthropometric assessment.**  At the beginning of the study, several anthropometric measurements were taken from the participants following the protocols described by Cabañas & Esparza [28]: i) *mass*; ii) *height*; iii) *sitting height*, defined as the distance between the vertex and the surface of the seat on which the participant sits; iv) *biileocrestal breadth* (width of the lower trunk), defined as the distance between the right and left anterior-superior iliac spine; v) *biacromial breadth* (upper trunk width), defined as the distance between the right and left acromial points; vi) *acromion-iliac index*, defined as the ratio between the biileocrestal and biacromial breadths multiplied by 100; and vii) *relative lower limb length*, defined

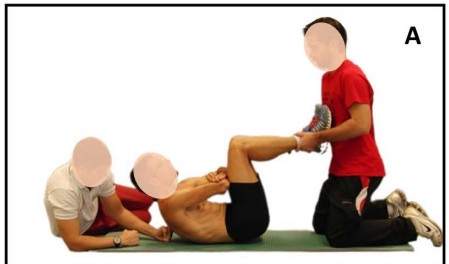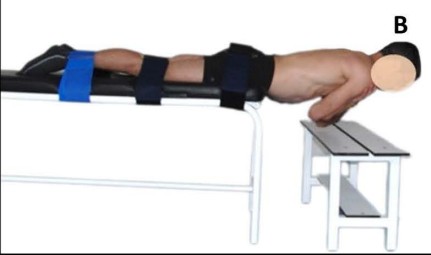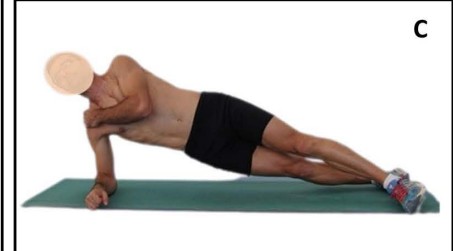

**Fig 1. A participant performing the isometric trunk muscle endurance field tests: A) Abdominal Ito test; B) Biering-Sorensen test; C) Right Side Bridge test.** Participants received verbal feedback each time they lost the required position in the tests and were encouraged to maintain the postures for as long as possible when they started to show signs of fatigue. The tests finished when the participants was not able to maintain the required position, and the duration of the tests (CASIO HS30W-N1V digital stopwatch) was registered as the result of the tests.

as the ratio between ileoespinal height (vertical height between the anterior-superior iliac spines and the support surface) and body height multiplied by 100.

## Statistical analysis

The normal distribution of all data series (Shapiro-Wilk test) and the possible existence of heteroscedasticity in each of the variables was analyzed. Subsequently, descriptive statistics [mean and standard deviation (SD)] were calculated for the trunk test scores and the anthropometric variables.

A two-way ANOVA was conducted to determine the differences between the two first testing sessions (learning effect) and sex: 3 (IT, BST, SBT) x 2 (session 1, session 2) x 2 (males, females). If significant differences were identified, the Bonferroni post hoc analysis was applied for pairwise comparisons.

The intra-rater relative reliability (participants rank order consistency from test to retest) was analyzed using the intra-class correlation coefficient ($ICC_{3,1}$), with its 95% confidence limit (95% CL). The qualitative interpretation of the ICC was: < 0.50, poor; 0.50–0.75, moderate; 0.75–0.90, good; > 0. 90, excellent [29]. The absolute reliability (consistency of repeated measurements for individuals across sessions) was determined by the change in mean (systematic error) and the typical error (TE) [24]. As data presented heteroscedasticity because the error was proportional participants´ score magnitude, TE was calculated as percentage following Hopkins´ procedure that log-transforms data using the following formula: $100 (e^s – 1)$, in which "s" is the standard error (SD of the difference between the first and the second session divided by $\sqrt{2}$) [24].

Multiple forward stepwise regressions were used to evaluate the relationship between the anthropometric variables and the trunk test scores with the data obtained in the third, fourth and fifth testing session. For this purpose, the estimation equation, the standard error of the estimation ($SE_{EST}$) and a correlation analysis (Pearson's coefficient) were calculated [30]. Only those variables that obtained a moderate-to-excellent level of relative reliability (i.e., ICC > 0.60) were used to perform Pearson correlation analyses (r) between them. To ensure the normality and homoscedasticity assumption in the correlational analyses, data (i.e., test scores and anthropometric variables) were logarithmic transformed.

The estimation equation was calculated as the equation obtained from the linear relationship between the logarithmically transformed values of the different tests and anthropometric parameters ($y = slope*X + intersection$). From this formula, the allometric parameters were calculated to normalize the results obtained in the different tests, using the following formula: $T/X^\alpha$, in which "T" is the result in the test, "X" the value of the anthropometric variable and "α" the slope of the linear relationship between log-transformed test scores and anthropometric parameters. The $SE_{EST}$ was calculated as the mean of the standard error of the difference between the values obtained in each of the tests for each of the participants and expressed as standard deviation. To interpret the $SE_{EST}$ values, Hopkins [31] suggests calculating the standardized

SE$_{EST}$ (SE$_{EST}$/ SD) of the criterion measure and then applying the following scale of values: < 0.1, trivial; 0.2–0.3, small; 0.3–0.6, moderate; 0.6–1.0, large; 1.0–2.0, very large; and >2.0, impractical.

The null hypothesis was rejected at the 95% significance level ($p \le .05$). Statistical analysis was performed with PASW statistics (version 27.0 for Windows 10; SPSS Inc., Chicago, IL, USA).

## Results

### Between-sex and reliability analyses

The ANOVA showed a significant effect in the within-subject factor "session" for the BST ($p \le .001$; F = 15.743) and the SBT ($p = .038$; F = 4.58), but not in the session*sex interaction in any of the tests ($p > .050$). Likewise, ANOVA found significant differences in the between-subject factor "sex" for the BST ($p = .021$; F = 5.741), with females showing higher values than males. Regarding reliability, the TE percentage ranged between 13.4% for the BST and 19.9% for the IT in males and between 10.2% for the BST and 30.0% for the IT in females. The ICC was greater than 0.80 in males and 0.71 in females in all tests (Table 1). Most anthropometric variables showed good or excellent absolute and relative reliability values (males: 3.4% > TE > 0.3%, ICC > 0.72; females: 3.8% > TE > 0.2%, ICC > 0.63). On the other hand, ANOVA showed a slight significant decrease (change in mean: −0.54%) in the males' mass between sessions ($p = .015$; F = 6.92) (Table 2).

The trunk endurance scores obtained for each of the tests (third, fourth and fifth session) were: 112.2 ± 67.2 s for the IT, 171.6 ± 62.4 s for the BST and 127.9 ± 44.1 s for the SBT in males and 125.6 ± 99.4 s for the IT, 219.6 ± 69.0 s for the BST and 117.4 ± 40.0 s for the SBT in females.

### Correlational analyses

Regarding the correlational results (Table 3), the IT scores showed a negative correlation with mass in males (r = −.436; $p = .026$) and females (r = −.562; $p = .019$) and with biileocrestal breadth (r = −.735; $p = .001$) and biacromial breadth (r = −.747; $p = .001$) in females. In addition, BST scores correlated negatively with biacromial breadth (r = −.379; $p = .050$) in males, and SBT scores showed a negative correlation with mass (r = −.703; $p < .001$), biileocrestal breadth (r = −.672; $p < .001$), biacromial breadth (r = −.601; $p = .001$) and iliac-acromion index (r = −.493; $p = .010$) in males and with relative lower limb length (r = −.493; $p = .038$) in females.

The software G*Power 3.1 (v3.1, University of Düsseldorf, Germany) was used to compute the post hoc achieved power for the relationship between anthropometric variables and trunk endurance tests scores with the following parameters: males = −0.379 < r < 0.703; α = 0.05; total sample size = 27 (calculations suggested a statistical power of 1–β = 0.637

**Table 1. Descriptive statistics and intra-rater relative and absolute reliability of the Ito test (IT), the Biering-Sorensen test (BST) and the Side Bridge test (SBT).**

| Variables | | Session 1 (mean±SD) | Session 2 (mean±SD) | %Change in mean (mean – 95% CL) | %Typical error (mean – 95% CL) | ICC $_{(3,1)}$ (mean – 95% CL) |
|---|---|---|---|---|---|---|
| IT (s) | Males | 90.9 ± 33.9 | 90.6 ± 43.6 | −2.5 (−12.0;7.9) | 19. 9 (15.4;28.2) | 0.80 (0.61;0.90) |
| | Females | 100.1 ± 60.7 | 105.2 ± 66.9 | 5.1 (−12.6;26.4) | 30.0 (21.8;48.2) | 0.71 (0.37;0.88) |
| | Global | 94.6 ± 46.1 | 96.5 ± 53.9 | 0.5 (−7.0;8.4) | 24.1 (20.3;30.1) | 0.75 (0.58;0.85) |
| BST (s) | Males | 143.4 ± 42.5 | 161.9 ± 52.2† | 12.4 (4.8;20.6) | 13.4 (10.4;18.8) | 0.84 (0.68;0.92) |
| | Females | 178.3 ± 42.5 | 193.7 ± 53.2 | 7.8 (0.7;15.4) | 10.2 (7.5;15.6) | 0.89 (0.72;0.96) |
| | Global | 157.4 ± 45.5 | 174.6 ± 54.3* | 10.6 (5.3;16.1) | 12.1 (10.0;15.6) | 0.87 (0.77;0.92) |
| SBT (s) | Males | 117.7 ± 36.5 | 122.5 ± 43.9 | 2.4 (−5.1;10.5) | 14.6 (11.3;20.5) | 0.87 (0.73;0.94) |
| | Females | 100.5 ± 38.4 | 109.6 ± 43.3 | 8.3 (−1.3;18.7) | 16.6 (12.7;24.3) | 0.87 (0.69;0.95) |
| | Global | 110.8 ± 37.8 | 117.3 ± 43.6* | 4.7 (−1.4;11.1) | 15.2 (12.4;19.5) | 0.86 (0.77;0.92) |

**Table 2. Descriptive statistics and intra-rater relative and absolute reliability of the anthropometric variables.**

| Variables | | Session 1 (mean±SD) | Session 2 (mean±SD) | %Change in mean (mean – 95% CL) | % Typical error (mean- 95% CL) | ICC $_{(3, 1)}$ (mean – 95% CL) |
|---|---|---|---|---|---|---|
| Mass (kg) | Males | 75.8±10.2 | 75.5±10.5*† | −0.5 (−0.9;-0.2) | 0.7 (0.6;0.9) | 1.00 (0.99;1.00) |
| | Females | 60.9±7.4 | 61.1±7.7 | 0.1 (−0.4;0.5) | 0.7 (0.6;1.0) | 1.00 (0.99;1.00) |
| Height (cm) | Males | 177.5±7.1 | 177.5±7.5 | 0.0 (−0.2;0.1) | 0.3 (0.2;0.4) | 1.00 (0.99;1.00) |
| | Females | 165.9±3.4 | 166.1±3.4 | 0.0 (−0.1;0.1) | 0.2 (0.2;0.3) | 0.99 (0.98;1.00) |
| Sitting height (cm) | Males | 146.6±3.4 | 146.7±3.3 | 0.1 (−0.1;0.4) | 0.6 (0.5;0.7) | 0.94 (0.89;0.97) |
| | Females | 141.8±2.6 | 141.0±2.1 | −0.3 (−0.8;0.2) | 0.9 (0.7;1.3) | 0.74 (0.48;0.88) |
| Biileocrestal breadth (cm) | Males | 28.9±2.2 | 28.5±2.4 | −1.4 (−2.8;0.0) | 2.9 (2.4;3.9) | 0.87 (0.76;0.94) |
| | Females | 27.6±1.6 | 27.7±1.4 | −0.1 (−1.8;1.6) | 2.7 (2.0;3.9) | 0.80 (0.57;0.92) |
| Biacromial breadth (cm) | Males | 41.9±1.4 | 41.8±1.7 | −0.4 (−1.3;0.6) | 1.9 (1.6;2.6) | 0.75 (0.55;0.87) |
| | Females | 37.9±1.5 | 37.5±1.4 | −1.1 (−2.4;0.2) | 2.1 (1.7;3.1) | 0.73 (0.45;0.88) |
| Acromion-iliac index (%) | Males | 69.0±4.4 | 68.3±4.8† | −1.1 (−2.7;0.6) | 3.4 (2.7;4.5) | 0.75 (0.56;0.87) |
| | Females | 73.1±3.1 | 74.3±3.0 | 1.3 (−0.3;3.0) | 2.6 (2.0;3.8) | 0.63 (0.30;0.83) |
| Relative lower limb length (%) | Males | 55.4±2.8 | 55.0±2.1 | −0.8 (−2.0;0.4) | 2.5 (2.0;3.3) | 0.72 (0.51;0.85) |
| | Females | 55.1±1.4 | 54.8±1.6 | −0.7 (−1.4;0.0) | 1.2 (0.9;1.7) | 0.83 (0.65;0.92) |

**Table 3. Correlations between the anthropometric variables and the Ito test (IT), the Biering-Sorensen test (BST) and the Side Bridge test (SBT).**

| | | IT | BST | SBT | Mass | Height | Sitting Height | RLLL | Biileocrestal Breadth | Biacromial Breadth | Iliac-acromion Index |
|---|---|---|---|---|---|---|---|---|---|---|---|
| IT | Males | – | .118 | .356 † | **−.436**\* | −.171 | −.191 | −.161 | −.275 | −.303 | −.170 |
| | Females | – | .246 | .233 | **−.562**\* | −.296 | −.460 † | −.110 | **−.735**\*\* | **−.747**\*\* | −.217 |
| BST | Males | – | – | .195 | −.227 | −.125 | −.223 | .058 | −.369 † | **−.379**\* | −.243 |
| | Females | – | – | .430 † | −.252 | −.136 | −.136 | .002 | −.096 | .265 | −.291 |
| SBT | Males | – | – | – | **−.703**\*\* | −.326 | −.140 | −.222 | **−.672**\*\* | **−.601**\*\* | **−.493**\* |
| | Females | – | – | – | −.302 | −.193 | .033 | **−.493**\* | −.204 | −.420 † | .201 |

RLLL: relative lower limb length;

*Significance: $p \leq .05$;

**Significance: $p \leq .001$; † Significance: $.05 < p < .10$.

to 0.996); females = 0.493 < r < 0.747; α = 0.05; total sample size = 18. (calculations suggested a statistical power of 1−β = 0.698 to 0.958).

Fig 2A shows the plot between the anthropometric variables (i.e., mass) and IT scores of original data for males, while Fig 3A shows the plot of the original data between anthropometric variables (i.e., biacromial breadth) and IT scores for females. After the log transformation of data (Figs 2B y 3B), the allometric parameter (the slope of the linear relationship between log-transformed test scores and anthropometric parameters) for mass in males was −1.479 with an $R^2 = 0.19$, while in females the allometric parameter was −9.3901 for the biacromial breadth, with an $R^2 = 0.53$. Normalization by allometric parameters allowed calculation of IT values avoiding the bias caused by anthropometric variables for males and females (Figs 2C and 3C): males: r = −.065; $p = .754$; females: r = −.04; $p = .862$.

Fig 4A shows the plot between the anthropometric variables (i.e., biacromial breadth) and the BST scores of the original data for males. After the log transformation of data (Fig 4B), the allometric parameter for the biacromial breadth and BST scores was −3.8288, with an $R^2 = 0.14$. Normalization allowed the effect of biacromial breadth to be removed from the BST scores in males (r = .029; $p = .884$) (Fig 4C). Linear regression analysis found no significant patterns for females.

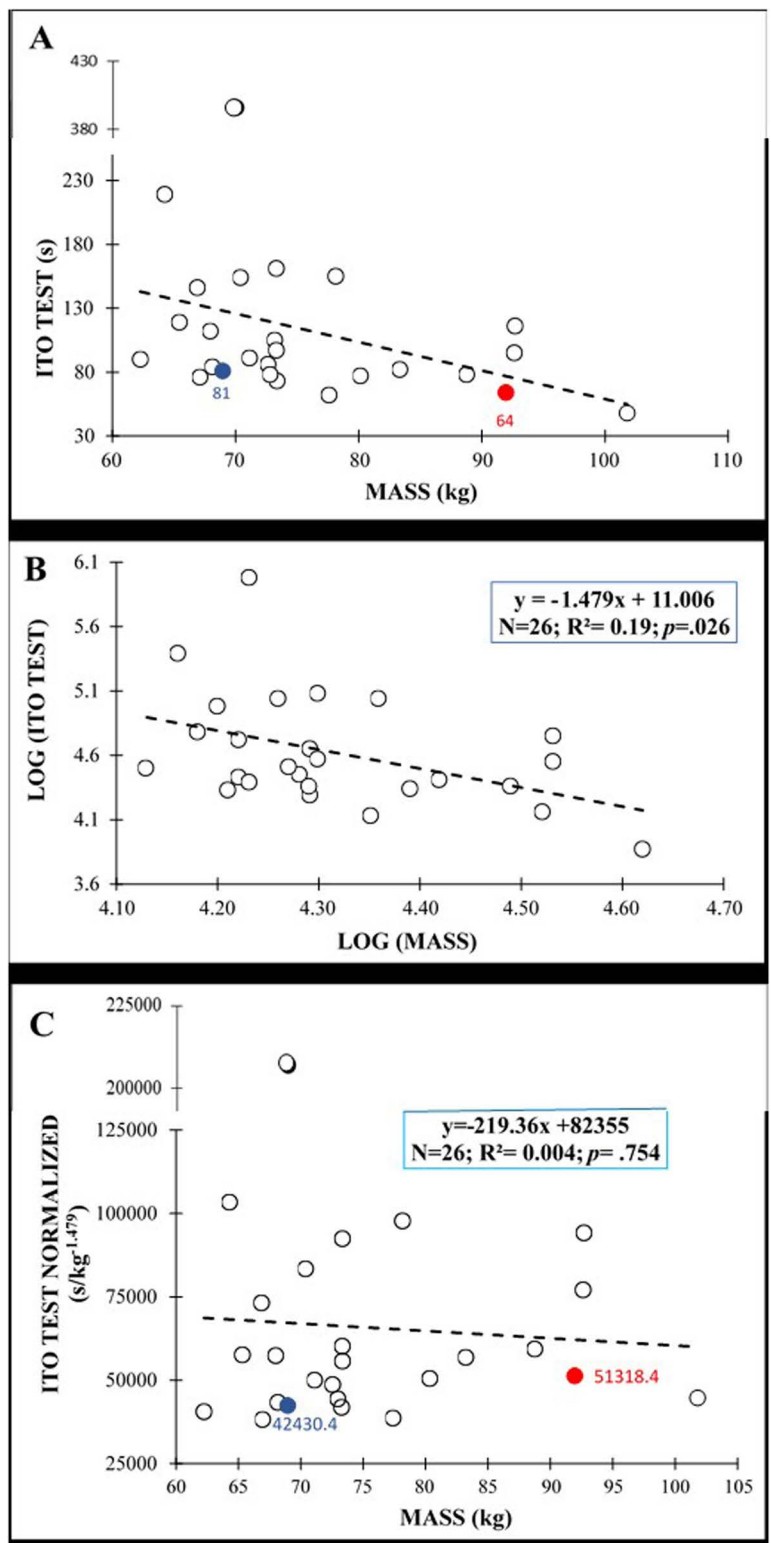

**Fig 2. Relationship between body mass and Ito test scores in males: (A) original data; (B) after logarithmic transformation to calculate the allometric parameter (slope of the regression line); (C) after allometric normalization.** The red and blue circles show the test scores of two participants before (figure A) and after the normalization (Fig C).

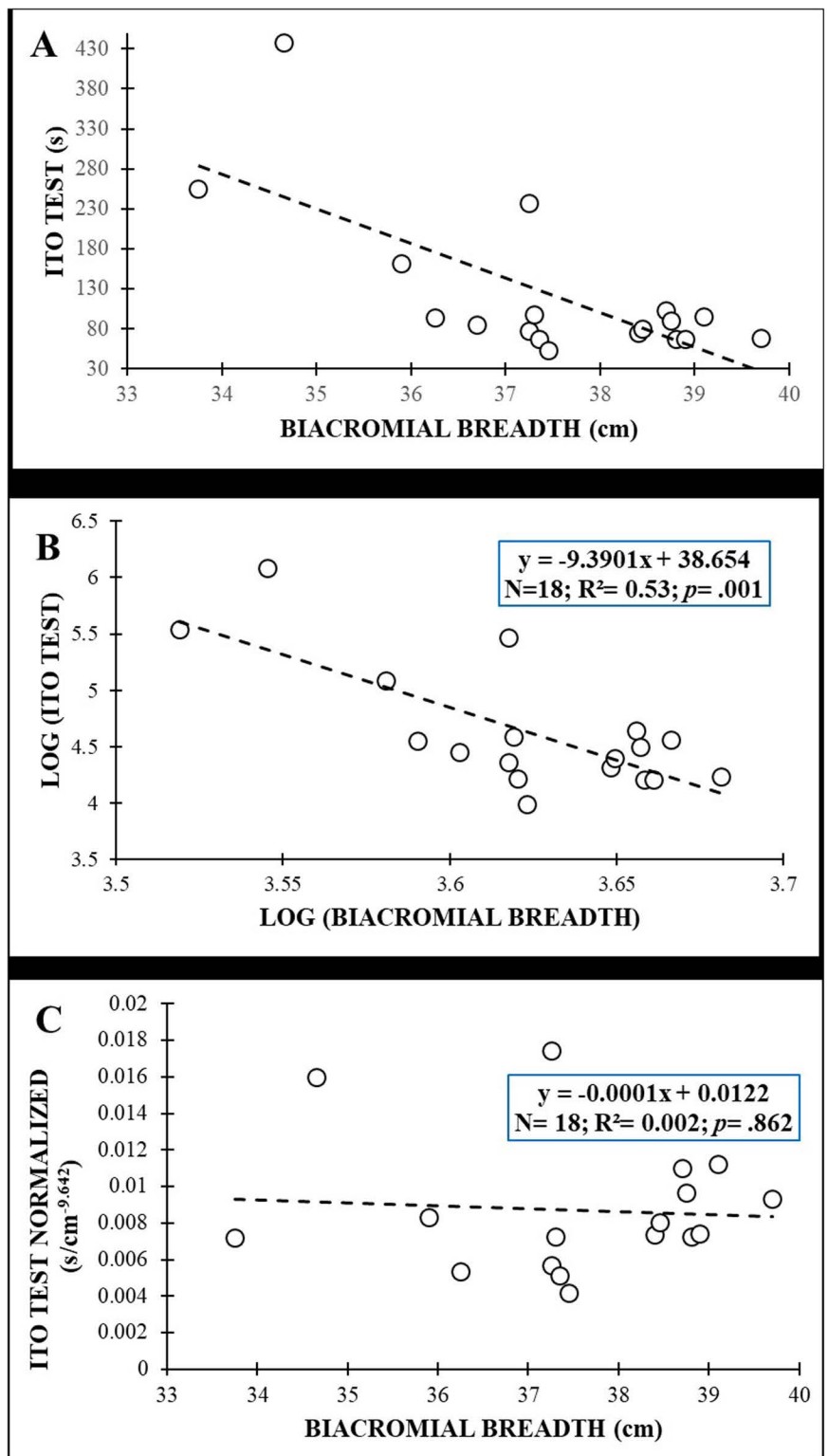

**Fig 3. Relationship between biacromial breadth and Ito test scores in females: (A) original data; (B) after logarithmic transformation to calculate the allometric parameter (slope of the regression line); (C) after allometric normalization.**

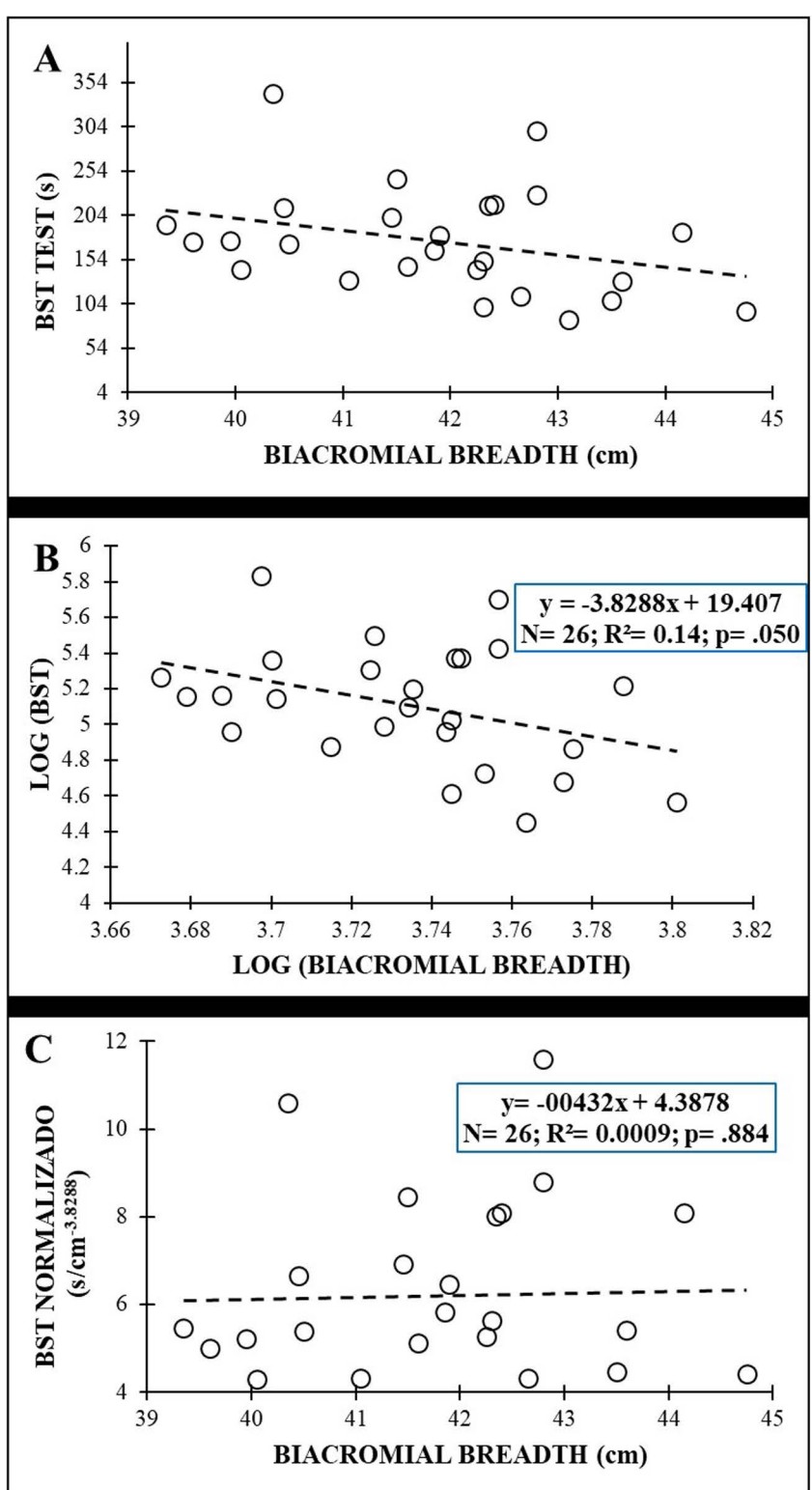

**Fig 4. Relationship between biacromial breadth and Biering-Sorensen test (BST) scores in males: (A) original data; (B) after logarithmic transformation to calculate the allometric parameter (slope of the regression line); (C) after allometric normalization.**

Fig 5A shows the plot of the original data between anthropometric variables (i.e., mass) and SBT scores for males while the Fig 6A shows the plot of original data between anthropometric variables (i.e., relative lower limb length) and SBT scores for females. After the log transformation of data, the allometric parameters were −1.843 for mass in males with an $R^2 = 0.50$, while in females the allometric parameter for the relative lower limb length was −8.273 with an $R^2 = 0.24$. Normalization allowed the effect of anthropometric variables to be removed from the SBT values for males (r = −.001; $p = .878$) (Fig 6A) and from the relative lower limb length for females (r = −.0032; $p = .822$) (Fig 6B).

Regarding the multiple regression analyses, the explained variance ranged from 14% for the BST (biacromial breadth) to 50% for the SBT (mass) in males and from 24% for the SBT (relative lower limb length) to 53% for the IT (biacromial breadth) in females. As shown in the Table 4, all the models showed a large standardized estimation error.

## Discussion

The use of isometric trunk holding tests (such as those analyzed in this study) is very common in fitness, rehabilitation, educational and research settings for assessing trunk muscular endurance. However, anthropometric characteristics affect the gravitational torque that participants need to counteract to hold the position and therefore they have an impact on test performance. Considering the lack of studies on the anthropometry effect on isomeric trunk endurance tests, the present study examined the relationship between several anthropometric measurements and BST, IT and SBT scores. In addition, test-retest reliability and the sex effect on test scores were also analyzed in order to improve knowledge about the characteristics of these tests and facilitate their use in different settings.

### Correlational analysis between the anthropometric measurements and the isometric trunk muscle endurance test scores

The data in the present study showed several significant correlations between the participants´ anthropometric characteristics and different test scores (Table 3). Concretely, the results showed a significant negative correlation between the body mass and IT scores in males and females (−.436 < r < −.562) and SBT scores in males (r = −.703), supporting previous negative correlations between the body mass or body mass index and trunk endurance test scores (BST: −0.29 < r < −0.39; SBT: −0.47 < r < −0.42; Plank test: r = −0.55; the 60º Flexion test: r: −0.47) [12–19]. A greater body mass implies an increased gravitational force which negatively affects trunk endurance. All these data confirm the importance of participants´ mass in the holding test performance, which has usually been ignored during the application of these tests in routine general practice [9,11,15,21–23].

Interestingly, other anthropometric variables related to body mass distribution and shape (the biacromial breadth, the biileocrestal breadth, the acromion-iliac index and the relative lower limb length) have also shown significant negative correlations with test performance in this study (Figs 2−6 and Table 3). These correlations (−.379 ≤ r ≤ −.747), although higher, are in line with those obtained by Dejanovic et al., [20] in children aged 7−14 years (boys: −.10 ≤ r ≤ −.26; girls: −.12 ≤ r ≤ −.34) and show the relevance of the body mass distribution and shape (i.e., upper-trunk mass, lower-trunk mass, limb length, etc.) on isometric trunk holding test performance. In this sense, a larger biacromial breadth can involve a larger upper-trunk mass, increasing the gravitational extension and flexion torques that participants have to counteract to hold the position during the IT and BST performance, respectively. In this sense, a trapezoidal trunk (i.e., an acromion-iliac index below 69.9) or short limbs (i.e., a relative length of the lower limbs below 54.9) [28], as is the case in our study (acromion-iliac index in males: 68.13±4.78; relative length of the lower limbs in females: 54.77±1.64), also represents a larger upper-trunk mass and a possible disadvantage for these holding test execution. In the SBT the upper-trunk mass has a single-upper limb support, so a larger biacromial breadth implies greater forces on the shoulder, elbow and forearm of the lowermost side and higher difficulty in maintaining the posture. In this sense, Juan-Recio et al., [17] showed that the shoulder muscle activation and the individuals' anthropometric characteristics played an important role in the SBT performance. In addition, during this test execution, the pelvis is approximately in the central part of the body mass that

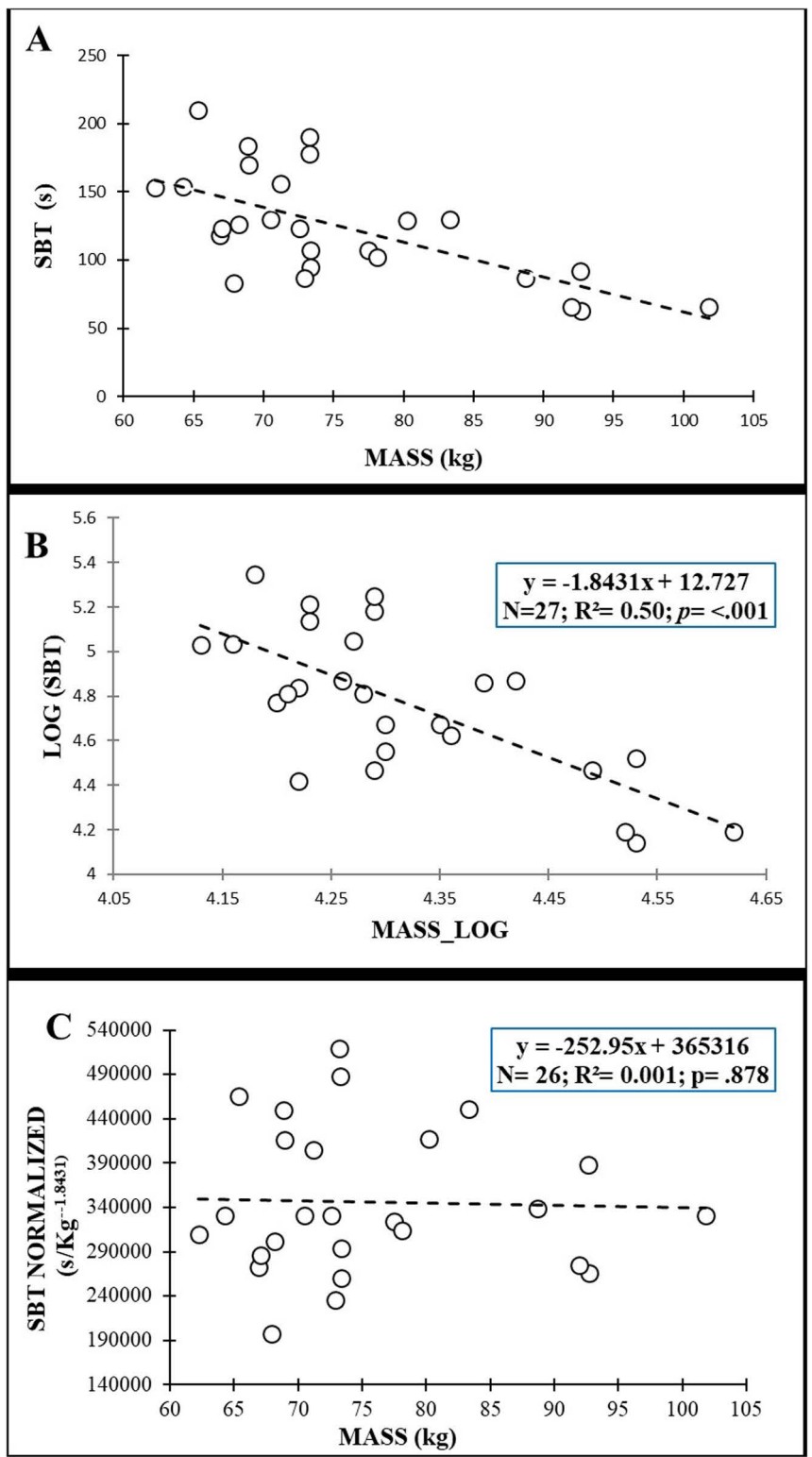

**Fig 5. Relationship of mass with Side Bridge test (SBT) scores in males: (A) original data; (B) after logarithmic transformation to calculate the allometric parameter (slope of the regression line); (C) after allometric normalization.**

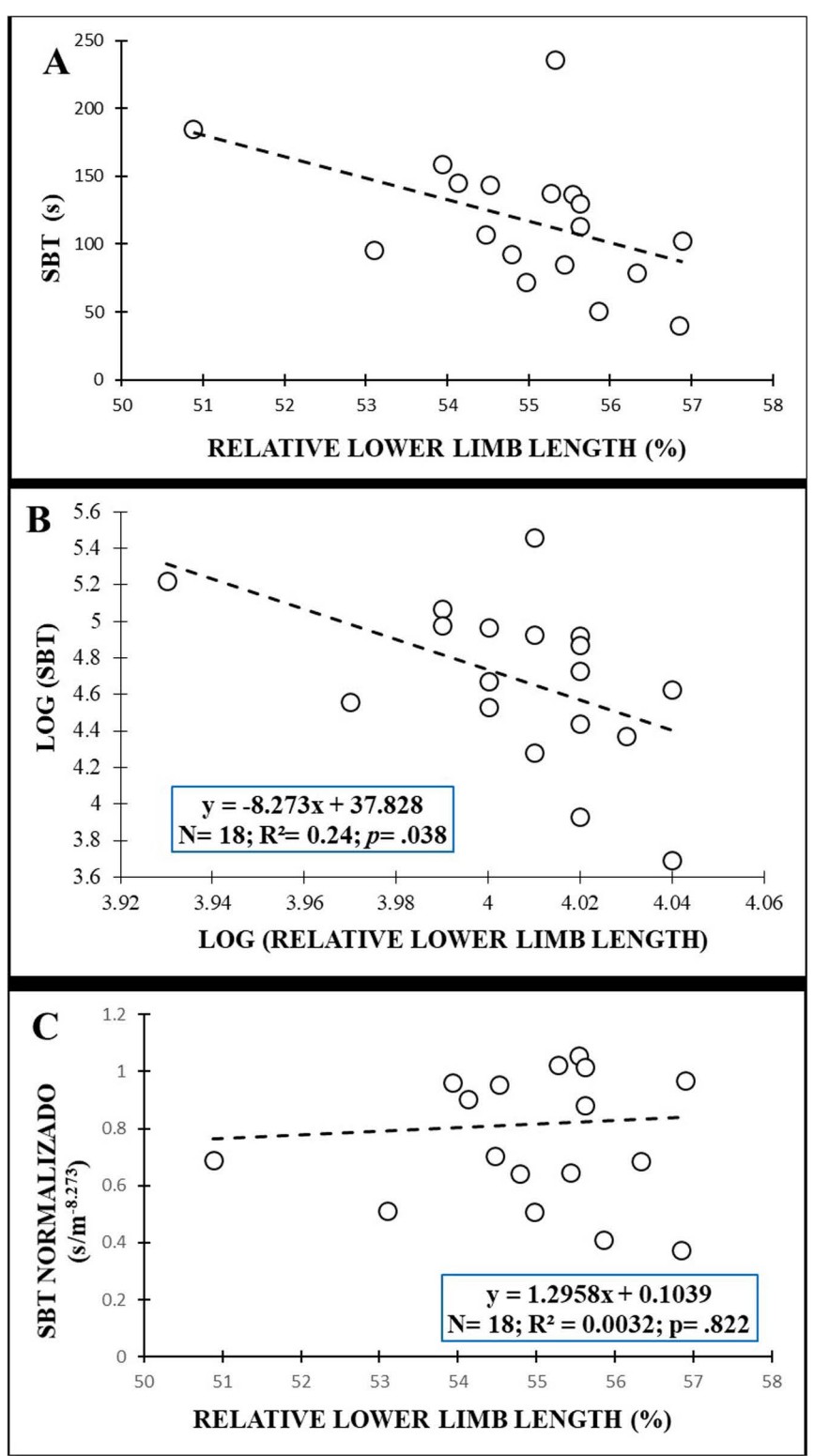

**Fig 6. Relationship between relative lower limb leg length and Side Bridge test (SBT) scores in females: (A) original data; (B) after logarithmic transformation to calculate the allometric parameter (slope of the regression line); (C) after allometric normalization.**

**Table 4. Parameters of the regression models for the Ito test (IT), the Biering-Sorensen test (BST) and the Side Bridge test (SBT).**

| Test | Sex | Predictive variables | α | 95% CL | TE[a] | R² | p |
|---|---|---|---|---|---|---|---|
| **IT** | Males | Mass | −1.479 | −2.75; −0.19 | 0.92 | 0.19 | .026 |
| | Females | Biacromial breadth | −9.3901 | −13.87; −4.21 | 0.73 | 0.53 | .001 |
| **BST** | Males | Biacromial breadth | −3.829 | −7.76; − 0.11 | 0.94 | 0.14 | .050 |
| | Females | – | – | – | – | – | – |
| **SBT** | Males | Mass | −1.843 | −2.62; −1.07 | 0.72 | 0.50 | <.001 |
| | Females | RLLL | −8.273 | −16.03; −0.51 | 0.90 | 0.24 | .038 |

α: allometric parameter; CL: confident limits; TE: Typical error of standarized estimation; R²: explained variance; RLLL: relative lower limb length.

must be held against gravity, therefore a larger biileocrestal breadth (i.e., a larger pelvis mass) also increases the gravitational torque and hinders test performance. Moreover, anthropometric differences such as a larger biileocrestal breadth may change muscle fiber directions and traction angles in the trunk, hip and/or shoulder joints, impacting the function of the muscles that are activated to counteract gravitational torque during the performance of these holding tests. Further research is needed to understand the effect of participants' anthropometric differences on muscle activation during these and other trunk muscle endurance tests, as well as the relationship of these changes in muscle function with test scores.

Given the large explained variance in several trunk endurance tests (14% < R² < 54%) showed by regression models, sport and health professionals should reflect what these tests actually measure. Considering that the small sample size prevents us for using the normalization algorithms presented in this study (Figs 3−6), from the authors´ point of view, raw trunk muscle endurance test scores should be used with caution when: i) analyzing the intervention effects in experimental studies in which the participants´ anthropometry may vary over time; ii) comparing individuals with significantly different anthropometric characteristics.

The findings of this study have some side potential implications on the performance of trunk stability exercises (such as side, front and back bridges) performed in similar positions than those used in this study. In this sense, the effect of the participants´ anthropometry on the gravitational torque that the participants must counteract to maintain the position during these exercises may have an impact on their intensity (i.e., the postural control challenge imposed on the individuals during the exercise execution) [32,33], reinforcing the need for a training load individualization when training with individuals with anthropometric differences.

### Reliability analysis of the isometric trunk muscle endurance test scores

The relative reliability was moderate-to-excellent for both, the isometric trunk muscle endurance test scores (0.71 ≤ ICC ≤ 0.89) and the anthropometrics measurements (0.63 ≤ ICC ≤ 1.00), which allowed all the variables to be used to perform the Pearson correlation analysis. The ICC values found for the IT, the SBT and the BST were similar to those obtained previously by other authors [9,25,26,34], demonstrating the ability of these tests to consistently discriminate differences in test performance between physically active individuals. On the other hand, as observed previously by Evans et al., [9], the TE values for the IT, the SBT and the BST were very high (12.1–24.1%), hindering their capacity to detect small changes in trunk performance over time (e.g., after an intervention or a detraining period), which could be especially problematic in highly trained individuals. Moreover, in line with previous studies on similar trunk tests [35,36], there was a significant between-session change (i.e., learning/repetition effect) in mean scores for the BST (10.6%) and the SBT (4.7%). This increase in test performance could be explained by different factors, for example: i) The ability to learn to make subtle postural adjustments during the test execution that could foster the alternation of activity between synergist muscles, reducing the impact of muscle fatigue; ii) improving tolerance to the sensations of muscle fatigue

(i.e., discomfort, stiffness, etc.) during test performance; and iii) the high motivation for improving test scores, common in athletes [35]. Overall, these data show the need to carry out extensive familiarization periods to reduce the learning effect of these tests, something that unfortunately does not usually occur in research, clinical and sport settings, in which only a test–retest or a single measurement is performed when evaluating trunk endurance [6,37].

### Between-sex analysis of the isometric trunk muscle endurance test scores

Regarding the between-sex analysis, females obtained significative higher scores than males in the BST (193.72 s *vs.* 161.93 s), supporting the results of previous studies [11,13,24,38]. Several hypotheses have been proposed to explain this sex-related difference in this test [39], although the most widely accepted theories are related to the greater upper body mass in males, the higher lumbar lordosis and the greater proportion of type I muscle fibers in females [40], and the differences in neuromuscular activation patterns (alternating activity between synergist muscles) [41]. On the other hand, although males scored better than females in the SBT (122.48 s *vs.* 109.61 s), the difference was not significant, which is contrary to the results of previous studies that did find significant differences [9,11,42]. Concerning the IT, females showed lager values than males (105.22 s *vs.* 90.63 s), which could be related to the fact that males had a significantly higher mass, sitting height, biacromial breadth, and iliac-acromion index than females (Table 2), and therefore higher gravitational extension torque during the test. However, these between-sex differences in the IT scores are not in line with the study of Ito et al., [26], and show the lack of agreement in scientific literature in regard to the sex-related differences in isometric trunk holding test performance. Further research is needed to explore the origin of these differences between studies, which could be related to differences in morphological characteristics, training preferences, sport practice, levels of motivation, sell-efficacy, etc. between participants.

### Study limitations

The findings are limited to a specific population of young university students. Moreover, since a larger sample would have allowed more generalizable results, future studies should perform correlation analysis with greater samples in different populations (e.g., sedentary people, high performance athletes, patients, older adults, etc.), which would allow to: i) to understand which anthropometric variables have a greater influence on the test scores better; ii) develop much more accurate predictive models and; and iii) elaborate normalized normative data that would allow to categorize the physical condition of the participants in a more realistic and adjusted way. Furthermore, considering the importance of body mass distribution in trunk muscle endurance performance, future studies should analyze the relationship between the masses of the relevant body parts (i.e., upper trunk, pelvis, etc.) together with body composition variables (i.e., fat mass and lean mass) and test scores.

## Conclusions

The present study provides important information for health and sport professionals about the IT, BST and SBT characteristics and limitations, which could improve their use and interpretation. The main findings of this study were the relationships of body mass and some anthropometric variables related to body mass distribution and shape (biacromial breadth, biileocrestal breadth, acromion-iliac index and relative lower limb length) with IT, BST and SBT performance, which must be considered when interpreting their results, especially in heterogenous populations. Based on the reliability analysis, these tests could rank participants consistently, but they require an extensive familiarization period and do not seem very accurate in detecting small changes in participants' performance over time. Finally, sex-related differences in trunk endurance scores should also be considered when using these tests in sport and clinical settings, especially in cases in which there are important anthropometric differences between males and females. Trainers and clinicians should consider individual anthropometric differences when interpreting test results, as larger body mass and upper body breadth may artificially lower endurance scores. Adjustments to normative values may be required in applied settings.

## Practical applications

Given the findings of this study, the authors recommend that sport and health professionals and researchers use the following equation to normalize trunk endurance tests (i.e., IT, BST and SBT) to anthropometric variables: $N = T/X^\alpha$, in which N = normalized test performance, T = test performance, X = anthropometric variable and α = allometric parameter. The importance of normalizing for anthropometric variables becomes even more evident when applied to a practical scenario such as comparing test scores among individuals with different morphological characteristics. An example can be observed in the Figs 2A and 2C, in which, the red and blue circles show the test scores obtained by two participants with quite different body mass (Blue participant: 69 kg and red participant: 92 kg) before and after allometric normalization. At first glance (Fig 2A), one might assume that the 92-kg individual performed worse (65 s) on the Ito test than the 69-kg individual (81 s). However, after normalizing the test scores using the allometric parameter (α = −1.479) (Fig 2C), the 92-kg individual ranks better than the 69-kg individual (normalized test performance = 51318.4 *vs*. 42430.4 s*kg). This is, the 92-kg individual has a better endurance capability allometrically adjusted by body mass. Furthermore, this anthropometric normalization would also be relevant when comparing pre- and post-test scores during those interventions in which changes in anthropometric variables as the body mass may occur.

## Author contributions

**Conceptualization:** CASTO JUAN RECIO, Francisco J. Vera-Garcia, Alejandro Lopez-Valenciano, David Barbado.

**Data curation:** CASTO JUAN RECIO, Francisco J. Vera-Garcia, Alejandro Lopez-Valenciano, David Barbado.

**Formal analysis:** Alejandro Lopez-Valenciano, David Barbado.

**Funding acquisition:** CASTO JUAN RECIO.

**Investigation:** CASTO JUAN RECIO, Francisco J. Vera-Garcia, Alejandro Lopez-Valenciano, David Barbado.

**Methodology:** CASTO JUAN RECIO, Francisco J. Vera-Garcia, Alejandro Lopez-Valenciano, David Barbado.

**Supervision:** CASTO JUAN RECIO, Francisco J. Vera-Garcia, Alejandro Lopez-Valenciano, David Barbado.

**Writing – original draft:** CASTO JUAN RECIO, Francisco J. Vera-Garcia, David Barbado.

**Writing – review & editing:** CASTO JUAN RECIO, Francisco J. Vera-Garcia, Alejandro Lopez-Valenciano, David Barbado.

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
