## [Decision Letter · Decision Letter 0]

2 Dec 2024

PONE-D-24-30908Anthropometric characteristics impact the participant’s performance in popular isometric trunk muscle endurance tests.PLOS ONE

Dear Dr. JUAN RECIO,

Thank you for submitting your manuscript to PLOS ONE. After careful consideration, we feel that it has merit but does not fully meet PLOS ONE’s publication criteria as it currently stands. Therefore, we invite you to submit a revised version of the manuscript that addresses the points raised during the review process.

We look forward to receiving your revised manuscript.

Kind regards,

Mário Espada, PhD

Academic Editor

PLOS ONE

“This research has been funded by the R+D+i for emerging research groups 2023 project of the Consellería de Educación, Universidades, y Empleo: CIGE/2022/ 22”

Reviewers' comments:

Reviewer's Responses to Questions

**Comments to the Author**

1. Is the manuscript technically sound, and do the data support the conclusions?

Reviewer #1: Yes

Reviewer #2: Partly

2. Has the statistical analysis been performed appropriately and rigorously? 

Reviewer #1: Yes

Reviewer #2: I Don't Know

3. Have the authors made all data underlying the findings in their manuscript fully available?

Reviewer #1: Yes

Reviewer #2: No

4. Is the manuscript presented in an intelligible fashion and written in standard English?

Reviewer #1: Yes

Reviewer #2: Yes

5. Review Comments to the Author

Reviewer #1: “Anthropometric characteristics impact the participant’s performance in popular isometric trunk muscle endurance tests”

The Authors are commended for completing a considerable amount of high-quality work. I think the results of this investigation will contribute to advance knowledge on trunk testing. In this perspective, this work has potential for becoming a foundational piece for future translational studies using these tests in patient populations.

Some minor comments are presented below for the Editor’s and Authors’ consideration.

INTRO

L85-87 I suggest rephrasing a bit the following “questions such as: Which anthropometric variables have a greater impact on the performance of these tests? Could anthropometric differences between sexes affect these test scores? etc.”

L97 The Authors may consider to:

1) State a specific hypothesis (and what they would anticipated as expected result from this investigation)

2) If and how more knowledge on this topic would benefit the scientific community and potential users of trunk control exercise testing

3) Who would care the most about new findings on this issue

METHODS

L100 and foll. The Authors may consider to better describe their cohort, e.g., how this sample was recruited, the level of physical activity, if they were from sampled from the general population or from university students’ populations, if there were athletes, etc. It is important to clarify the sample under study, also for replicability purposes

L100 forty-five participants: was sampling stopped because you reached a predefined size of the cohort? Any a priori power analyses predefining an expected difference? Please, clarify

DISCUSSION

Overall, the discussion read well even though in some instances it includes results or data repetition which I would suggest deleting.

L343 “Between-sex analysis of the isometric trunk muscle endurance test scores”. I truly appreciated this paragraph, which discussed the findings in a physiologically driven manner. I wonder whether the previous sections would benefit from more physiological rather than biomechanic-only reasoning

L377 The manuscript needs to be re-read carefully for some minor grammar errors and typos, such as “specially” at this line.

Study limitations

I would suggest running a post hoc power analysis to see how much power you put together and if the sample is really limited as you acknowledged in the limitations.

Conclusions

I wonder whether this section could also include some elaboration on the potential clinical relevance of these tests and study findings for clinicians and their clients.

Thank you for the opportunity to revise this fine paper.

Reviewer #2: The authors have submitted a manuscript in which they try to link anthropometric values with test performance of three different, but widely used trunk muscle endurance tests. Additionally the investigation also contains reliability data as one part of the investigation.

Major

Part one (reliability investigation)

For me it's somehow surprising that the anthropometric values at least partly were less reliable than the values for the tests. This does not speak for exact anthropometric measures, therefore questioning the whole investigation. Question: what anthropometric values were used for the correlation analyses?

What do the authors mean when using the term "relative reliability?

In this part values for the tests are provided – where can the reader find the values used for the second part?

The provided data are ALL in contrast to the originally published values. This requires explanation.

As the Ito test originally contains a flexion and an extension part, this should be mentioned and explained. The only hint is provided in figure 1.

Part two (correlation calculations)

Again: what are the test data used for the calculations. Further: which anthropometric data were used? Why do the authors only provide logarithmic data? As no original data were provided, the reviewer has no chance to get an idea about the original dataset.

In the manuscript it is mentioned several times, that especially the upper body weight has impact on trunk muscle endurance test. Why did the authors refrain from also using this particular value? Further, the used anthropometric values are all measures in the frontal plane. But humans come as three-dimensional subjects. For me it is inconsequent to only use such parameters and not also parameters of "deepness" i.e. in the sagittal plane or, as already recommended, masses of relevant body parts.

Detailed

L37-38

Not clear how many tests were performed in the mentioned sessions.

L40

What do the authors mean by using the wording "relative reliability"?

L52-53

The statement about familiarization is not supported by the results

L60

"increasing …. endurance" sounds strange as no intervention is mentioned

L93-94

Absolute and relative reliability – not explained

L102

Please provide the kind of physical activity, as this might have impact on the endurance capacity

L120-121

Eight minutes break between endurance tasks to exhaustion sound very short. How about cumulating fatigue for the three tests?

L124-126

No criterion mentioned to identify learning effects. Provocative question: do you think that for a simple holding test e learning effect can occur?

L128

Please be consistent in citation style

L130 – figure 1

Please provide the subfigures in the same order as described in the text. Also: just mention, that the Ito test consists of two tests: one for the abdominals and one for the back muscles. obviously here the abdominal test was used.

For the SBT slightly different times were reported per side - which side was tested here?

L143

Please define the ileospinal height (start – end)

L155

Again: relative reliability… unclear.

L180-181

Scale of values unclear (also after reading the Hopkins paper)

Results section

Please provide a clear structure, best by using subheadings – first reliability analysis and then the correlation results.

Table 1

Already mentioned: as the provided test times are considerably different from the original articles these data require some explanation.

The term " Significant between-xxx differences" is simply wrong

Table 2

Please provide all distance values in cm. You provide relative differences for all anthropometric variables together with typical errors and ICC values. As the reliability data are partly worse than the test reliability, this calls for an explanation.

Table 3

What is the Iliac-acromion Index? Not explained.

Figures 2, 3, 4

x- slope and increments seem all to be wrong. But maybe the data are related to the "original" but not provided data.

Figure 5

two linear interpolations but only one equation.... accounts for both subfigures

L268-270, L293, L308-309, L347-350

Please see my comment about other promising anthropometric variables..

L294-295

You state "can" - right. But be clear that a possibility is not a causality.

L298

not really, since the resulting body slope will increase together with this measure, consequently reducing the load (following the sine function). Here, body length would have been expected to be negatively correlated with holding time.

L304

What is the relationship between biileocrestal breadth and lower-trunk mass? Calls for explanation.

L339-340

Why do you think your results contain a learning effect? This calls for a hypothesis.

L353-358

All mentioned influencing factors were present also in the original Ito investigation. So this is no explanation.

L380

In all original articles data were provided separately for both sexes….

---

## [Author Response · Author response to Decision Letter 1]

29 Jan 2025

The authors would like to thank the Reviewers and Editors for their advice and recommendations. We believe that the manuscript is stronger as a result of their comments.

The authors’ responses to each comment are bolded, and changes in the manuscript are presented in red.

Reviewer #1: “Anthropometric characteristics impact the participant’s performance in popular isometric trunk muscle endurance tests”

The Authors are commended for completing a considerable amount of high-quality work. I think the results of this investigation will contribute to advance knowledge on trunk testing. In this perspective, this work has potential for becoming a foundational piece for future translational studies using these tests in patient populations.

Some minor comments are presented below for the Editor’s and Authors’ consideration.

INTRO

L85-87 I suggest rephrasing a bit the following “questions such as: Which anthropometric variables have a greater impact on the performance of these tests? Could anthropometric differences between sexes affect these test scores? etc.”

We have rephrasing the sentence… “including which anthropometric variables have a greater impact on test performance and the potential impact of sex-based differences in test scores”

L97 The Authors may consider to:

1) State a specific hypothesis (and what they would anticipated as expected result from this investigation)

2) If and how more knowledge on this topic would benefit the scientific community and potential users of trunk control exercise testing

3) Who would care the most about new findings on this issue

We have included this sentence at the end of the introduction “Based on previous studies (12–18), individuals´ anthropometric characteristics will have an impact on the isometric trunk muscle endurance test performance. These findings could improve the use and interpretation of these trunk muscle endurance field tests in clinical and sport settings”.

METHODS

L100 and foll. The Authors may consider to better describe their cohort, e.g., how this sample was recruited, the level of physical activity, if they were from sampled from the general population or from university students’ populations, if there were athletes, etc. It is important to clarify the sample under study, also for replicability purpose.

We have better describe our cohort including information about the type of population and physical activity performed “… with a regular practice of 2-5 times a week of 60-120 min per day of moderate physical activity for a total of 120-300 min per week voluntarily participated in the study. The participants were recruited from a university student´ population and were recreative athletes (i.e. soccer, gymnastics, basketball, running, etc).”

L100 forty-five participants: was sampling stopped because you reached a predefined size of the cohort? Any a priori power analyses predefining an expected difference? Please, clarify

Authors: We did not calculate a priori power analysis. Following, the reviewer´s suggestion we have included a post hoc power analysis.

DISCUSSION

Overall, the discussion read well even though in some instances it includes results or data repetition which I would suggest deleting.

L343 “Between-sex analysis of the isometric trunk muscle endurance test scores”. I truly appreciated this paragraph, which discussed the findings in a physiologically driven manner. I wonder whether the previous sections would benefit from more physiological rather than biomechanic-only reasoning

Authors: For the discussion of the data we have tried to use both results from previous studies and biomechanical principles and criteria. Whenever we have considered it appropriate, we have discussed the results in a more physiological manner. In response to your comment, in the section Correlational analysis between the anthropometric measurements and the isometric trunk muscle endurance test scores, we have included some sentences where we interpret some results based on the differences in muscle morphology and function:

“Moreover, anthropometric differences, such as a larger biileocrestal breadth, may change muscle fiber directions and traction angles in the trunk, hip and/or shoulder joints, impacting the function of the muscles that are activated to counteract gravitational torque during the performance of these holding tests. Further research is needed to understand the effect of participants’ anthropometric differences on muscle activation during these and other trunk muscle endurance tests, as well as the relationship of these changes in muscle function with test scores.”

L377 The manuscript needs to be re-read carefully for some minor grammar errors and typos, such as “specially” at this line.

Authors: The manuscript has been submitted to Proof-Reading-Service.com for editing and proofreading.

Study limitations

I would suggest running a post hoc power analysis to see how much power you put together and if the sample is really limited as you acknowledged in the limitations.

A post-hoc power analysis has been performed and included in manuscript.

The software G*Power 3.1 (v3.1, University of Düsseldorf, Germany) was used to compute the pos hoc achieve power for the relationship between anthropometric variables and trunk endurance tests scores with the following parameters: males= -0.379 <r< 0.703; α= 0.05; total sample size= 27. Calculations suggested a statistical power of 1–β= 0.637 to 0.996; females= 0.493 <r< 0.747; α= 0.05; total sample size= 18. Calculations suggested a statistical power of 1–β= 0.698 to 0.958.

The sample size is a limitation to elaborate normalized normative data that would allow to categorize the physical condition of the participants in a more realistic and adjusted way as we indicated in limitation section, but in general not for statistical analysis.

Conclusions

I wonder whether this section could also include some elaboration on the potential clinical relevance of these tests and study findings for clinicians and their clients.

Authors: We have included a practical application section which shows an example of two participants and the implications of our results for trainers and clinicians.

Thank you for the opportunity to revise this fine paper.

Reviewer #2: The authors have submitted a manuscript in which they try to link anthropometric values with test performance of three different, but widely used trunk muscle endurance tests. Additionally the investigation also contains reliability data as one part of the investigation.

Major

Part one (reliability investigation)

For me it's somehow surprising that the anthropometric values at least partly were less reliable than the values for the tests. This does not speak for exact anthropometric measures, therefore questioning the whole investigation. Question: what anthropometric values were used for the correlation analyses?

Authors: As we indicated in page 8; line 168: “Only those variables that obtained a moderate-to-excellent level of relative reliability (i.e., ICC > 0.60) were used to perform Pearson correlation analyses (r) between them”

We have revised our results and some reliability data of anthropometric variables were wrong. We agree with you that it can be somehow surprising that the anthropometric values at least partly were less reliable than the values for the tests and it could be also surprising for the readers. In addition, we appreciate your comment because it is really that it is not well explained in the manuscript or can be a bit confusing and therefore you and the readers could think that we do not have measured the anthropometric characteristics in an accurate way. But this is not true.

These are a brief definition for relative and absolute reliability. The relative reliability represents how well the rank order of participants in one trial is replicated in the second trial (normally evaluated with the ICC) while the absolute reliability is the degree to which repeated measurements vary for individuals (normally evaluated with the SEM).

If we analyze the reliability of the biacromial breadth in females we show that the SEM is 2.1% and ICC=0.73. This means that the error is only 0.75 cm for someone whose biacromial breadth is 37.5 cm (the females sample mean). If the sample is very homogeneous (as in our case), the ICC could be low because it is very difficult to replicate the same rank order between sessions, because the small SEM (0.75 cm) can do that participants change the rank order.

The logarithmic transformation of data was used in the reliability analysis to reduce the bias derived from the non-uniformity of the error and also in the correlation analysis to ensure the normality and homoscedasticity assumption.

The data of reliability of the table 1 and 2 refer to original data shown in the tables.

What do the authors mean when using the term "relative reliability?

- The relative reliability represents the consistency of rank order of participants from test to retest, while the absolute reliability is the consistency of repeated measurements for individuals across sessions.

Hopkins WG. Measures of reliability in sports medicine and science. Sports Medicine. 2000;30:1–15.

We have included a brief explanation of absolute and relative reliability in methods section.

In this part values for the tests are provided – where can the reader find the values used for the second part?

The logarithmic transformation is a powerful technique in statistics and data analysis that helps stabilize variance and normalize distributions.

Again, we appreciate your comment because it is really that it is not well explained in the manuscript or can be a bit confusing. We have included the figures with the original data and we have modified the text in the manuscript explaining which is the purpose of log transformation and what is the allometry parameter (slope of the straight line of the regression with log data).

Now, the readers can compare the original data with the normalized data and see how the anthropometric variables influence on test performance. We have included an example in one of the plots and a practical application section to explain it and its implications for practitioners and health and sport profesionals.

The provided data are ALL in contrast to the originally published values. This requires explanation.

-In the section Between-sex analysis of the isometric trunk muscle endurance test scores, at the Discussion section, we showed that BST and SBT values are in line with previous originally published data except for the significant differences between males and females obtained previously in SBT. No significant differences can be related with the population characteristics, sample size, etc.

-Concerning the IT, which between-sex differences scores are not in line with the study of Ito et al. (25), the authors hypothesizes that they could be related to the fact that males had significantly higher mass, sitting height, biacromial breadth, and/or iliac-acromion index than females (table 2), and therefore higher gravitational extension torque during the test.

As the Ito test originally contains a flexion and an extension part, this should be mentioned and explained. The only hint is provided in figure 1.

- The specific Ito test used in this study has been mentioned in the manuscript:

“The BST (24), the IT (25) and the SBT (11) were performed to asses extensor, flexor and lateral flexor trunk endurance, respectively as described by Juan-Recio et al., (2014)”.

In addition, we have mentioned it in the introduction

“…In order to answer these questions and facilitate the use and interpretation of some of the most representative isometric trunk muscle endurance field tests [i.e., Biering-Sorensen test (BST), Abdominal Ito test (IT), Side Bridge test (SBT)]…”

Part two (correlation calculations)

Again: what are the test data used for the calculations. Further: which anthropometric data were used? Why do the authors only provide logarithmic data? As no original data were provided, the reviewer has no chance to get an idea about the original dataset.

The data used in the correlation analysis were the same as those used in the reliability analysis. In both analysis, data were log-transformed for both tests performance and anthropometric characteristics to ensure the homoscedasticity assumption.

In the manuscript it is mentioned several times, that especially the upper body weight has impact on trunk muscle endurance test. Why did the authors refrain from also using this particular value? Further, the used anthropometric values are all measures in the frontal plane. But humans come as three-dimensional subjects. For me it is inconsequent to only use such parameters and not also parameters of "deepness" i.e. in the sagittal plane or, as already recommended, masses of relevant body parts.

As you mention, an individual’s body mass and its distribution can impact performance in the isometric trunk muscle endurance test. We appreciate your suggestion and agree that it would have been better to use the actual masses of the relevant body parts. However, these measurements require an indirect and more complex calculation compared to the anthropometric measurements commonly used in this study and in previous research. (Dejanovic et al., 2012, 2014). Due to the complexity of calculating the masses of body parts, we have used anthropometric measurements related to these masses, for example: greater biacromial and biileocrestal diameters imply greater masses in the upper trunk and pelvis, respectively.

We have included this in the limitation section

Furthermore, considering the importance of body mass distribution in trunk muscle endurance performance, future studies should analyze the relationship between the masses of the relevant body parts and test scores.

Detailed

L37-38

Not clear how many tests were performed in the mentioned sessions.

The number of tests performed in each session have been clarify in the text:

“… performed the three isometric trunk holding tests twice in two testing sessions to perform the reliability analysis and later, the three tests were performed once more, but in different sessions (one for each test)…”

L40

What do the authors mean by using the wording "relative reliability"?

- The relative reliability represents how well the rank order of participants in one trial is replicated in the second trial and normally is expressed by ICC. We have included a brief explanation of absolute and relative reliability in Methods section.

We have included a brief explanation of absolute and relative reliability in methods section.

L52-53

The statement about familiarization is not supported by the results

- We have specified that some of the tests require a longer familiarization period.

“… but some of them require an extensive familiarization period”.

L60

"increasing …. endurance" sounds strange as no intervention is mentioned

-We have eliminated the word “increasing” and we have rewritten the sentence introducing “…the development of trunk muscle endurance…”

L93-94

Absolute and relative reliability – not explained

We have included a brief explanation of absolute and relative reliability in Methods section. In addition, the sentences include the cite of Hopkins´ study in which both are explained.

Hopkins WG. Measures of reliability in sports medicine and science. Sports Medicine. 2000;30:1–15.

We have included a brief explanation of absolute and relative reliability in methods section.

L102

Please provide the kind of physical activity, as this might have impact on the endurance capacity

Some examples of the kind of physical activity have been provided.

L120-121

Eight minutes break between endurance tasks to exhaustion sound very short. How about cumulating fatigue for the three tests?

Although rest periods of 5-10 min have been used in previous studies (Evans et al., 2007; Mcgill et al., 1999), to mitigate the effect of fatigue, the three tests were first performed in a counterbalanced way for reliability analysis, and later, the three tests were performed in separate sessions for correlation analysis.

L124-126

No criterion mentioned to identify learning effects. Provocative question: do you think that for a

---

## [Decision Letter · Decision Letter 1]

18 Mar 2025

PONE-D-24-30908R1Anthropometric characteristics impact the participant’s performance in popular isometric trunk muscle endurance tests.PLOS ONE

Dear Dr. JUAN RECIO,

Thank you for submitting your manuscript to PLOS ONE. After careful consideration, we feel that it has merit but does not fully meet PLOS ONE’s publication criteria as it currently stands. Therefore, we invite you to submit a revised version of the manuscript that addresses the points raised during the review process.

We look forward to receiving your revised manuscript.

Kind regards,

Mário Espada, PhD

Academic Editor

PLOS ONE

Journal Requirements:

Additional Editor Comments:

Dear Authos,

Please revise the manuscript consider the feedback by the reviewers,

Thank you.

Best regards.

Reviewers' comments:

Reviewer's Responses to Questions

**Comments to the Author**

1. If the authors have adequately addressed your comments raised in a previous round of review and you feel that this manuscript is now acceptable for publication, you may indicate that here to bypass the “Comments to the Author” section, enter your conflict of interest statement in the “Confidential to Editor” section, and submit your "Accept" recommendation.

Reviewer #3: All comments have been addressed

Reviewer #4: (No Response)

Reviewer #5: All comments have been addressed

2. Is the manuscript technically sound, and do the data support the conclusions?

Reviewer #3: Yes

Reviewer #4: Partly

Reviewer #5: Yes

3. Has the statistical analysis been performed appropriately and rigorously? 

Reviewer #3: Yes

Reviewer #4: N/A

Reviewer #5: Yes

4. Have the authors made all data underlying the findings in their manuscript fully available?

Reviewer #3: Yes

Reviewer #4: No

Reviewer #5: No

5. Is the manuscript presented in an intelligible fashion and written in standard English?

Reviewer #3: No

Reviewer #4: Yes

Reviewer #5: Yes

6. Review Comments to the Author

Reviewer #3: Please check the apostrophes and some contractions where they have grammatical errors. The article is very interesting and addresses important aspects regarding public health, however, I recommend justifying how the results could be generalized having a small population; Likewise, expand the influence of sex on the results.

Reviewer #4: Dear Authors,

Thank you for submitting your manuscript, "Anthropometric Characteristics Impact the Participant’s Performance in Popular Isometric Trunk Muscle Endurance Tests," to PLOS ONE. Your study provides valuable insights into the relationship between anthropometric characteristics and trunk muscle endurance test performance. The research is well-structured and has potential implications for sports science, rehabilitation, and clinical assessments.

However, after a thorough review, I have identified several areas that require substantial revisions before the manuscript can be considered for publication. Specifically, the study lacks explicit hypotheses, a priori power analysis for sample size justification, and a clear explanation of certain methodological choices, such as rest periods and statistical transformations. Additionally, the discussion should further elaborate on the physiological mechanisms underlying the observed correlations and their practical applications in training and clinical settings. While the study is generally well-written, minor grammatical and clarity issues should also be addressed.

I have provided detailed comments and recommendations to strengthen your manuscript. If these concerns are adequately addressed, I believe your study could make a significant contribution to the field. I look forward to your revised version.

Best regards,

Reviewer #5: This is a simple but useful study. The authors reviewed three widely used tests for assessing core strength. Most users do not consider that the anthropometric characteristics of the subjects could influence the test results. Based on this review, the use of the tests should take into account individual differences in body composition and sex. This constitutes an interesting contribution to practice, as the tests are a useful tool for measuring core strength economically without requiring complicated and expensive equipment. However, the study demonstrates that greater caution is warranted in their use, and users should keep in mind that the tests have low sensitivity, so changes with training should also be analyzed with caution.

The statistical analyses are original, useful, and comprehensive. They provide a broad perspective on the reliability and relationship with anthropometric measurements.

The authors do not present the database.

7. PLOS authors have the option to publish the peer review history of their article (what does this mean? ). If published, this will include your full peer review and any attached files.

**Do you want your identity to be public for this peer review?** For information about this choice, including consent withdrawal, please see our Privacy Policy .

Reviewer #3: No

Reviewer #4: No

Reviewer #5: No

---

## [Author Response · Author response to Decision Letter 2]

16 Apr 2025

The authors would like to thank the Reviewers and Editors for their advice and recommendations. We believe that the manuscript is stronger as a result of their comments.

The authors’ responses to each comment are bolded, and changes in the manuscript are presented in red.

Review of the Manuscript: Anthropometric Characteristics Impact the Participant’s Performance in Popular Isometric Trunk Muscle Endurance Tests

General Evaluation

The manuscript investigates the relationship between anthropometric characteristics and performance in isometric trunk muscle endurance tests, a relevant topic in sports sciences and rehabilitation. The study is well-structured, and the methodology is clear. However, several issues require revision before publication. These include missing details on sample selection, statistical power, and methodological explanations. Furthermore, the discussion needs stronger connections to practical applications, and some statistical choices require further justification. Below is a detailed breakdown of these concerns.

1. Title and Abstract

Title Revision

• The current title is clear but could be more specific in highlighting both reliability and performance analysis. A suggestion: "The Impact of Anthropometric Characteristics on Isometric Trunk Muscle Endurance Tests: A Reliability and Performance Analysis"

• This version clarifies the dual focus on reliability and performance.

Authors: We have changed the title in the manuscript

The Impact of Anthropometric Characteristics on Isometric Trunk Muscle Endurance Tests: A Reliability and Performance Analysis

Abstract (Lines 31-54)

• The abstract effectively summarizes the study but lacks a strong emphasis on practical applications. Consider adding a concluding statement addressing how the findings can be utilized in sports science, rehabilitation, and clinical settings.

• The phrase "some of them require an extensive familiarization period" (Line 53) should be revised to indicate which tests specifically require longer familiarization.

Authors: We have indicated which tests require a specifically longer familiarization period and we have included a concluding statement addressing how the findings can be utilized in sports science, rehabilitation, and clinical settings.

“In conclusion, trainers and clinicians should consider individual anthropometric and sex differences when interpreting test results, as a larger body mass and upper body breadth may artificially lower endurance scores. Adjustments to normative values may be required in applied settings. Moreover, based on the reliability analysis, these tests could be used to classify participants consistently, but the BST and the SBT require an extensive familiarization period”

• The methodological details about the log transformation applied to statistical data should be mentioned briefly to avoid confusion later in the text.

Authors: We have mentioned the methodological details about the log transformation applied to statistical data briefly.

To ensure the normality and homoscedasticity assumption, data (i.e., test scores and anthropometric variables) were logarithmically transformed.

2. Introduction

Lack of Explicit Hypotheses (Line 97)

• While the study’s objectives are outlined, explicit hypotheses are missing. The manuscript should clearly state:

1. "We hypothesize that individuals with greater body mass and broader anthropometric dimensions will exhibit lower endurance test performance due to increased gravitational torque."

2. "Sex-based differences will influence test performance, with females potentially showing higher endurance in certain tests due to differences in muscle fiber composition and fat distribution."

• This addition enhances clarity and scientific rigor.

Authors: We have included both hypothesis in the manuscript.

“Based on previous studies (12–18), we hypothesize that individuals with greater body mass and broader anthropometric dimensions will exhibit lower endurance test performance due to increased gravitational torque. In addition, sex-based differences will influence test performance, with females potentially showing higher endurance in certain tests due to differences in muscle fiber composition and fat distribution.”

Recent Literature Missing

• The introduction references study up to 2018, but more recent literature should be incorporated to support the research gap. Specifically:

o Recent systematic reviews (2020-present) on core endurance and its relation to anthropometric factors should be discussed.

o If there are any recent studies evaluating plank or Biering-Sorensen tests in specific populations, they should be included.

Authors: We have included two recent cites which related BMI with the Biering-Sorensen test and the 60º flexion test, but despite our best efforts we have not found any recent systematic reviews (2020-present) on core endurance and its relation to anthropometric factors in scientific literature.

Aydoğan K, Kostanoğlu A, Törpü GC. Effect of Body Mass Index on Balance, Trunk Muscle Endurance, Functional Mobility and, Physical Activity in College Students. Bezmialem Sci. 2024;12(4):441–9.

Vlažná D, Krkoška P, Kuhn M, Dosbaba F, Batalik L, Vlčková E, et al. Assessment of Lumbar Extensor Muscles in the Context of Trunk Function, a Pilot Study in Healthy Individuals. Appl Sci 2021;11(20):9518.

3. Methods

Sample Description (Lines 100-107)

• The manuscript states that participants were "physically active individuals," but this description is vague.

• Key missing details:

o Were the participants trained athletes, recreational athletes, or untrained individuals?

o What was the criterion for being considered "physically active"? (e.g., self-reported activity levels, validated scales like IPAQ)

o Was the sample size predetermined through power analysis? (Only a post hoc analysis is provided.)

• Suggested revision:

o "Participants were recruited based on self-reported physical activity of at least 120–300 minutes per week of moderate exercise, but no structured core training."

Authors: We have changed the participants’ description in the abstract.

“Forty-five recreational athletes (27 males and 18 females) performed the three isometric trunk holding tests”

In the participants section, we included that participants were recreational athletes and that they completed a questionnaire about their medical and athletic history to evaluate their health status and regular physical activity.

“They were recreational athletes (i.e. practicing soccer, gymnastics, basketball, running, etc.), with a regular practice of 60-120 min of moderate to intense physical activity 2-5 times a week (for a total of 120-300 min per week, approximately). They completed a questionnaire about their medical and athletic history to evaluate their health status and regular physical activity.”

Testing Procedures (Lines 120-121)

• The 8-minute rest period between endurance tasks is questionable.

o Is this rest period sufficient to prevent residual fatigue from influencing subsequent test performance?

o Previous studies have used 10–15 minutes of rest in similar protocols.

o It would be beneficial to cite literature justifying the chosen rest duration.

Authors: We agree with the reviewer. Although rest periods of 5-10 min have been used in previous studies (Chan et al., 2005; Evans et al., 2007; McGill et al., 1999), the three tests were performed first in a counterbalanced way to mitigate the effect of fatigue in our study for reliability analysis, and later, the three tests were performed in separate sessions for correlation analysis.

-Chan RH. Endurance times of trunk muscles in male intercollegiate rowers in Hong Kong. Arch Phys Med Rehabil. 2005 Oct;86(10):2009-12. doi: 10.1016/j.apmr.2005.04.007. PMID: 16213246.

- Evans K, Refshauge KM, Adams R. Trunk muscle endurance tests: reliability, and gender differences in athletes. J Sci Med Sport. 2007 Dec;10(6):447-55. doi: 10.1016/j.jsams.2006.09.003. Epub 2006 Dec 1. PMID: 17141568.

-McGill SM, Childs A, Liebenson C. Endurance times for low back stabilization exercises: clinical targets for testing and training from a normal database. Arch Phys Med Rehabil. 1999 Aug;80(8):941-4. doi: 10.1016/s0003-9993(99)90087-4. PMID: 10453772.

Authors: We have included these references in the manuscript when we describe the rest period.

Reliability of Anthropometric Measurements (Line 155)

• The manuscript does not report inter-rater or intra-rater reliability for anthropometric assessments.

o Were measurements taken by the same investigator?

o If not, was reliability assessed between raters?

o These details are crucial to ensuring measurement consistency.

Authors: The measurements were taken by the same investigator: The intra-rater relative reliability of anthropometric measurements were reported in tables 1 and 2. We have specified this in the table caption and the statistical analysis.

4. Results

Learning Effect and Test Repetitions (Lines 339-340)

• The study suggests a learning effect for BST and SBT. However, no comparative analysis is presented to confirm this.

o A comparison of first vs. last trial performances across all tests should be included to validate the learning effect hypothesis.

o Were participants given additional instructions before later trials that could have influenced their improvement?

Authors: This paragraph has been included in statistical analysis section.

“A two-way ANOVA was conducted to determine the differences between the two first testing sessions (learning effect) and sex: 3 (IT, BST, SBT) x 2 (session 1, session 2) x 2 (males, females). If significant differences were identified, the Bonferroni post hoc analysis was applied for pairwise comparisons.”

We analyzed the reliability in the first two sessions in which all the tests were performed together. The examiner gave the same instructions during both sessions and the participants were encouraged to maintain the postures for as long as possible when they started to show signs of fatigue.

Statistical Power and Sample Size Justification (Lines 272-274)

• A post hoc power analysis is provided, but no a priori power analysis justifying the initial sample size is presented.

o Without an a priori power analysis, it is unclear if the sample size was adequate to detect meaningful effects.

o The manuscript should clarify:

How was the number of 45 participants determined?

Would a larger sample have produced more generalizable results?

Authors: We carried out a priori power analysis considering a r= -0.55 (Nuzzo and Mayer, 2013) and an α= 0.05; 1–β= 0.80. The total sample size calculated was 16 participants (32 participants considering males and females).

Because a few correlations obtained in this study were under 0.55 we considered more appropriate that including a post-hoc power analysis in order to show the real statistical power of this study was more appropriate. In the limitation section, we suggested that future studies should perform correlation analysis with larger samples in different populations:

Study limitations

These findings are limited to a specific population of young university students. Moreover, since a larger sample would have allowed more generalizable results, future studies should perform correlational analysis with larger samples in different populations (e.g., sedentary people, high performance athletes, patients, older adults, etc.), which would allow: i) to understand which anthropometric variables have a greater influence on the test scores better; ii) to develop much more accurate predictive models and; and iii) to elaborate normalized normative data that would allow to categorize the physical condition of the participants in a more realistic and adjusted way.

Correlation Analysis Interpretation (Table 3, Lines 210-218)

• The negative correlations between body mass and endurance scores are well discussed, but the physiological mechanisms need further explanation:

o Why does body mass negatively affect endurance? Is it due to increased gravitational force or metabolic inefficiency?

o Why does the acromion-iliac index correlate with lower endurance? What biomechanical factors contribute to this?

• The discussion should incorporate muscle function, biomechanical load distribution, and physiological adaptations.

Authors: We have included the explanation why body mass negatively affect endurance

“A greater body mass implies an increased gravitational force which negatively affects trunk endurance”

In the discussion section we discussed why the acromion-iliac index correlates with lower endurance

“In this sense, a trapezoidal trunk (i.e. an acromion-iliac index below 69.9) or short limbs (i.e. a relative length of the lower limbs below 54.9) (25), as is the case in our study (acromion-iliac index in males: 68.13±4.78; relative length of the lower limbs in females: 54.77±1.64), also represent a larger upper-trunk mass and a possible disadvantage for these holding test execution. In the SBT the upper-trunk mass has a single-upper limb support, so a larger biacromial breadth implies greater forces on the shoulder, elbow and forearm of the lowermost side and higher difficulty in maintaining the posture. In this sense, Juan-Recio et al., (17) showed that the shoulder muscle activation and the individuals’ anthropometric characteristics played an important role in the SBT performance”.

5. Discussion and Conclusion

Real-World Application of Findings (Lines 353-358)

• The conclusion should provide a clearer practical takeaway for trainers, clinicians, and researchers.

• For example:

o How can sports scientists apply this research in athletic screening?

o How should clinicians interpret endurance test results for rehabilitation patients?

• Suggested addition: "Trainers and clinicians should consider individual anthropometric differences when interpreting test results, as larger body mass and upper body breadth may artificially lower endurance scores. Adjustments to normative values may be required in applied settings."

Limitations and Future Directions

• The manuscript lacks an explicit mention of key limitations:

o Generalizability: Findings are limited to a specific population of young, university students.

o Body Composition Variables: Fat mass and lean mass, which could influence endurance performance, were not measured.

o No Direct Measurement of Muscle Activation: Surface EMG or other biomechanical analyses could enhance understanding.

• The authors should propose future research directions, such as:

o Investigating how specific training adaptations affect endurance test results.

o Examining endurance performance in older adults or elite athletes.

Authors: We have included the suggestions of the reviewer in the study limitations section.

6. Language and Writing

Grammar and Clarity

• The manuscript is generally well-written but contains minor language errors.

• Example corrections:

o Line 377: "specially" → should be "especially."

o Line 60: "increasing endurance" → should be "developing endurance" (as no intervention was conducted).

o Line 52-53: The claim that tests require "extensive familiarization" is unsupported by results and should be rewritten as:

"Our results suggest that some of these tests may require longer familiarization for accurate performance assessment."

• Though the manuscript has been proofread, a final language revision is recommended to improve clarity and precision.

Authors: We have revised all language errors and we have changed “increasing” by “development”.

Considering the fact that trunk muscle endurance deficits and imbalances have been related with low back pain (1–3) and that trunk muscle fatigue has a harmful effect in muscular coordination, postural control and spine stability (4,5), the development of trunk muscle endurance is a common training program goal for injury treatment and prevention, functional capacity improvement in daily tasks and sport performance (6–8).

As we stated in the statistical analysis an ANOVA was conducted to determine the differences between the two first testing sessions. In addition, in the results section we indicate that “The ANOVA showed a significant effect in the within-subject factor “session” for

---

## [Editor Report · Decision Letter 2]

2 May 2025

Anthropometric characteristics impact the participant’s performance in popular isometric trunk muscle endurance tests.

PONE-D-24-30908R2

Dear Dr. CASTO JUAN RECIO,

We’re pleased to inform you that your manuscript has been judged scientifically suitable for publication and will be formally accepted for publication once it meets all outstanding technical requirements.

Kind regards,

Mário Espada, PhD

Academic Editor

PLOS ONE

---

## [Editor Report · Acceptance letter]

PONE-D-24-30908R2

PLOS ONE

Dear Dr. JUAN RECIO,

I'm pleased to inform you that your manuscript has been deemed suitable for publication in PLOS ONE. Congratulations! Your manuscript is now being handed over to our production team.

Kind regards,

on behalf of

Dr. PLOS Manuscript Reassignment

Staff Editor

PLOS ONE